# Successful Treatment of an Esophago-Tracheobronchial Fistula Using Double Stenting to Correct Initial Stent Migration: A Case Report and Literature Review

**DOI:** 10.3390/jcm13237382

**Published:** 2024-12-04

**Authors:** Yuya Nishio, Hideki Matsuo, Shinsaku Nagamatsu, Kazuki Shioya, Chisa Yamamoto, Shoma Kikukawa, Kyohei Matsuura, Yuki Fujimoto, Masakazu Uejima, Kei Moriya

**Affiliations:** 1Department of Gastroenterology, Nara Prefecture General Medical Center, Nara 630-8581, Japan; m12082yn@jichi.ac.jp (Y.N.); h.matsuo@nara-hp.jp (H.M.); joe.montana.no16@gmail.com (S.N.); kairiki_moririn@yahoo.co.jp (K.S.); nmu114112cy@gmail.com (C.Y.); ks50205020@gmail.com (S.K.); graynurseshark07@rakuten.jp (K.M.); v7df792@yahoo.co.jp (Y.F.); 2Department of Endocrinology and Metabolism, Nara Prefecture General Medical Center, Nara 630-8581, Japan; m3m16d333@yahoo.co.jp

**Keywords:** metallic stent, esophageal cancer, esophago-tracheobronchial fistula, stent migration, double stenting technique

## Abstract

**Background**: Esophago-tracheobronchial fistula is a severe and often fatal complication in patients with advanced esophageal cancer, requiring prompt attention. The standard treatment involves the placement of a covered stent, which is relatively simple to perform and effectively seals the fistula. However, stent migration remains a common issue, highlighting the need for improved methods to prevent it. **Case Presentation**: We developed an innovative double stenting method approach utilizing two types of metal stents for cases where conventional stenting led to early stent dislodgement. This technique combines the benefits of uncovered and fully covered stents while minimizing their limitations. Furthermore, it is straightforward and adaptable. In two cases treated at our facility, this method successfully maintained complete fistula coverage until the patients’ deaths, allowing them to consume food orally. **Conclusions**: Here, we describe the procedure in detail and discuss its significance, as our findings demonstrate the effectiveness of the double stenting technique.

## 1. Background

As advanced esophageal cancer progresses, various complications are common, with esophago-tracheobronchial fistula occurring in approximately 5–15% of cases [1,2] and leading to severe, rapidly worsening conditions [3,4]. The placement of a covered stent in the fistula is regarded as an effective treatment for esophago-tracheobronchial fistula, as it is relatively simple to perform and reliably seals the fistula [5]. However, stent placement for fistula closure has a relatively high rate of stent migration [6], indicating a need for techniques that prevent such displacement. Although specific methods to minimize stent migration have been proposed, the migration rate remains significant at around 68–87%, leaving room for further improvement [7].

This study presents a case in which a “double stenting method”, using two metal stents of the same size (one uncovered and one fully covered), effectively prevented stent migration in a patient with advanced esophageal cancer who experienced early stent migration after the conventional stenting of an esophago-tracheobronchial fistula.

## 2. Case Presentations

Case 1. An 82-year-old woman visited her local doctor after experiencing difficulty swallowing and weight loss for about a month. Her medical history included type 2 diabetes, hypertension, and osteoporosis, for which she was taking sitagliptin, olmesartan, and eldecalcitol. She did not drink alcohol or smoke and had no prior history of cancer. Upper gastrointestinal endoscopy revealed a 3/4 circumscribed raised tumor with central depression in the middle and lower esophagus (28–35 cm from the incisors), and histopathology confirmed squamous cell carcinoma. A whole-body computed tomography (CT) scan showed thickening of the esophageal mucosa in the middle and lower sections, along with enlarged right cervical and upper thoracic paraesophageal lymph nodes, but no distant metastases. The patient was diagnosed with advanced thoracic esophageal cancer, classified as cT3N1M0, stage IIIa under the cTNM system. She was admitted to our hospital due to a sudden fever just before the scheduled start of her first chemoradiotherapy session. On examination, she measured 148 cm in height, weighed 42 kg, had a body temperature of 37.7 °C, a pulse rate of 93/min, blood pressure of 134/65 mmHg, respiratory rate of 16 breaths/min, and an SpO_2_ of 95%. Blood tests indicated an elevated white blood cell count with a shift to the left (WBC 13,300/µL, neutrophils 83.2%) and a high CRP (C-reactive protein) level of 8.82 mg/dL, suggesting acute infection. A computed tomography (CT) scan revealed a new heterogeneous soft shadow near the primary esophageal lesion, highly suggestive of a mediastinal abscess linked to esophageal perforation (Figure 1). Antibiotic therapy (ABPC/SBT) was immediately initiated, and the patient was placed on a fasting regimen. On the fifth day, a fully covered esophageal stent (OD 18 mm, length 15 cm, HANAROSTENT^®^ by Boston Scientific, Boston, MA, USA) was placed to close the fistula (Figure 2A). However, by the 13th day, the stent had dislodged into the stomach (Figure 2B).

The stent was retrieved orally by using forceps to tighten the lasso at its tip under endoscopic guidance. Given the risk of recurrent migration with fully or partially covered metal stents, a double stenting approach was adopted as a new method for fistula closure. The procedure began with the placement of an uncovered metal stent (Niti-S^®^ by Taewoong medical Co., Ltd., Seoul, Republic of Korea) with an 18 mm outer diameter and a 12 cm length over the tumor, covering the fistula area (Figure 3A). Two days later, once the tumor had grown into the stent and stabilized it (Figure 3B), a fully covered metal stent of the same dimensions (HANAROSTENT^®^ by Boston Scientific) was inserted inside the initial stent, creating a double layer at the same position (Figure 3C). Both stents were securely fixed at the proximal end with hemostatic clips (Figure 3D).

She was able to start a liquid diet on the 20th day, and the mediastinal abscess remained stable, allowing her to continue oral intake. The quantity and content of the diet were gradually increased, and on the 40th day, she was able to consume a normal diet. With the increase in food intake, her body weight increased from 42 kg (on admission) to 45.5 kg and her nutritional status improved. Following the initiation of radiotherapy for esophageal cancer on the 23rd day, no stent migration or worsening of the mediastinal abscess were observed, and she was discharged on the 43rd day after a short recovery. During the 108 days before her eventual death from cancer, no stent migration or blockage occurred.

Case 2. A 71-year-old man visited our hospital after experiencing discomfort upon swallowing for about a month. He had no medical history but had a beer drinking history of one liter every day and smoking history (20 pieces per day for 30 years). Upper gastrointestinal endoscopy revealed advanced esophageal squamous cell carcinoma (cT3N2M0 cStage III) in the mid-thoracic esophagus (30–35 cm from the incisors), and neoadjuvant DCF chemotherapy (docetaxel, cisplatin, and 5-fluorouracil) was performed. During the course of treatment, the patient became aware that they had a fever and persistent cough, and the inflammatory findings of blood tests and CT of the chest were most suspicious for a mediastinal abscess associated with esophageal perforation (Appendix A). After admission, he was fasting and receiving antimicrobials, and a full-coverage esophageal stent (HA-NAROSTENT^®^) with an outer diameter (OD) of 18 mm and a length of 12 cm was implanted against the tumor area for fistula closure on the third day, but it was found that the stent had fallen out in the stomach on the ninth day (Appendix A). To avoid the possibility of stent redislodgement, an uncovered stent (Niti-S^®^), 18 mm OD and 12 cm long, was placed in the tumor with the fistula on the 16th day, and a covered stent (Niti-S^®^) of the same size was placed inside it one week later to create a complete double-layer structure; the mouth ends of both stents were fixed with clips (Appendix A). Esophagogram with Gastrografin^®^ contrast on the 26th day showed no contrast leak (Appendix A), and oral intake was resumed. Radiotherapy and chemotherapy for esophageal cancer were started on the 31st and 43rd day, respectively, but there was no stent deviation or worsening of mediastinal abscess, and the patient was discharged on the 48th day. Thereafter, there were no stent deviations or occlusions during the 216 days until death.

## 3. Discussion and Conclusions

Esophago-tracheobronchial fistulas are relatively common in advanced esophageal cancer. These fistulas impair daily life by making oral intake difficult, leading to poor nutritional status and persistent respiratory infections, which often prove fatal [8]. Treatment options for closing esophago-tracheobronchial fistulas include cyanoacrylate sealing of the fistula opening [9], filling with fibrin glue [10], coverage with polyglycolic acid (PGA) sheets [11], clip-based suture techniques such as “over-the-scope clipping” (OTSC) [12,13], and the use of covered stents [14,15].

Both cyanoacrylate filling and fibrin glue filling are suitable only for patients with a small fistula diameter [9,10]. While PGA sheets can be used for relatively large fistulas, the success rate is lower for fistulas compared to perforations [16]. In contrast, OTSC is applicable to larger fistulas; however, its effectiveness on hard tissues like fistulas is reported to be under 60% [13]. It can be assumed that the success rate would be even lower for bronchial fistulas associated with the full-layer invasion of esophageal cancer. In contrast, the placement of a covered stent for esophago-tracheobronchial fistulas can be performed minimally invasively under endoscopic guidance [4,17] and is an effective treatment expected to alleviate respiratory symptoms through fistula closure. Of course, except endoscopic procedure, surgical approach might be considered as a treatment method of esophago-tracheobronchial fistulas. However, cases of fistula formation are basically classified as advanced stage, and the general condition of these patients is often poor. In these cases, the esophagus is tightly adhered to the bronchus and lungs due to tumor invasion, making it difficult to close the fistula by surgical dissection, which is a physically invasive procedure. Consequently, in many medical facilities, the standard first-line treatment for esophago-tracheobronchial fistulas related to advanced esophageal cancer with deep invasion is the use of covered stents.

There are two types of covered stents used for fistula closure: fully covered and partially covered. Fully covered stents do not cause stenosis due to tumor growth within the stent lumen, but they have a lower coefficient of friction than partially covered stents, making them more prone to displacement from the original implantation site. However, partially covered stents are thought to be less likely to deviate (Table 1), although some studies suggest no significant difference in deviation rates between the two types [18]. In a review of 47 esophageal stenting procedures conducted at our institution over a 10-year period from November 2013 to October 2023, stents were placed to cover a fistula in 17 cases. Among these, no stent migrations were observed in patients with highly stenotic esophageal lumens at the fistula site, which made endoscope passage difficult (*n* = 14). However, in the remaining cases (*n* = 3), where endoscope passage was adequate and there was no significant luminal stenosis, stent migration occurred in two cases (66%), both involving fully covered stents.

We typically select fully covered metal stents for esophageal stenting due to a past case of acute respiratory failure caused by severe tracheal compression from the opening of an implanted stent. Fortunately, we were able to perform the emergency stent removal without significant complications, likely because it was a fully covered stent. Therefore, the decision regarding the type of indwelling stent should consider multiple factors.

In general, when stenting is performed to dilate stenotic lesions, the stent is securely anchored by its radial force. However, patients needing fistula coverage may not have significant stenosis, and in cases of fistulas with mild stenosis, the stent’s radial force may not effectively secure it, increasing the likelihood of stent migration. This means that even if some risk of stent migration is anticipated, it is crucial to ensure that the covered stent is firmly positioned at the fistula site, as fistula coverage is essential. While OTSC has been reported as a method to secure the stent to the esophageal mucosa to prevent migration, there have been case reports indicating that deviations can still occur with this technique [19,20]. Furthermore, OTSC implantation is technically complex, and the necessary equipment can be costly.

We developed the “double stenting method” as a new approach for the easy and reliable placement and fixation of stents in cases of migration. This method involves placing a covered stent inside an uncovered stent. The first step is to position an anchor stent (uncovered) at the fistula site. After a few days, tumor growth is observed within the stent, which helps secure it firmly to the fistula. Then, a fully covered stent of the same diameter and length is inserted into the uncovered stent, creating a stent-in-stent configuration. The uncovered stent and the covered stent are held together by horizontal frictional forces. To further prevent deviation, the mouth ends of the two overlapping stents are secured with hemostatic metal clips, ensuring complete coverage of the fistula site for an extended period. The two cases of the “double stenting method” performed at our hospital have remained free of stent migration for 108 days and 216 days, respectively, until the patients’ deaths from the primary disease, while maintaining the coverage of the fistula area and allowing for safe oral intake.

The success of this method in preventing stent migration relies not just on the horizontal frictional force between the two metal stents but also on enhancing the pull-out resistance of the inner stent. This is achieved by aligning the flared ends of both the fully covered and uncovered stents so that their ends fit together. For this alignment to work effectively, the diameter and stent length of both stents must be the same, making it easier to grasp and secure the mouth edges with metal clips. The “double stenting method” does not require specialized techniques, which is a significant advantage. However, a limitation is the high cost of the metal stents used. In Japan, medical insurance typically covers only one stent for malignant esophageal stricture or esophago-tracheobronchial fistula. Currently, the uncovered stent costs USD 745, and the fully covered stent costs USD 845, making them expensive. Nevertheless, given reports of cases that needed multiple stents due to recurrent deviations [21], the simple and reliable “double stenting method” holds considerable significance. It is anticipated that further case accumulation will demonstrate the effectiveness and reliability of this method, potentially leading to insurance coverage for the use of two stents.

In conclusion, the “double stenting method”, which utilizes both an uncovered and a fully covered stent, is a straightforward and effective approach for stenting in cases where there is a deviation of a stent implanted for esophago-tracheobronchial fistula due to deep invasion by esophageal cancer, making it highly beneficial in clinical practice.

## Figures and Tables

**Figure 1 jcm-13-07382-f001:**
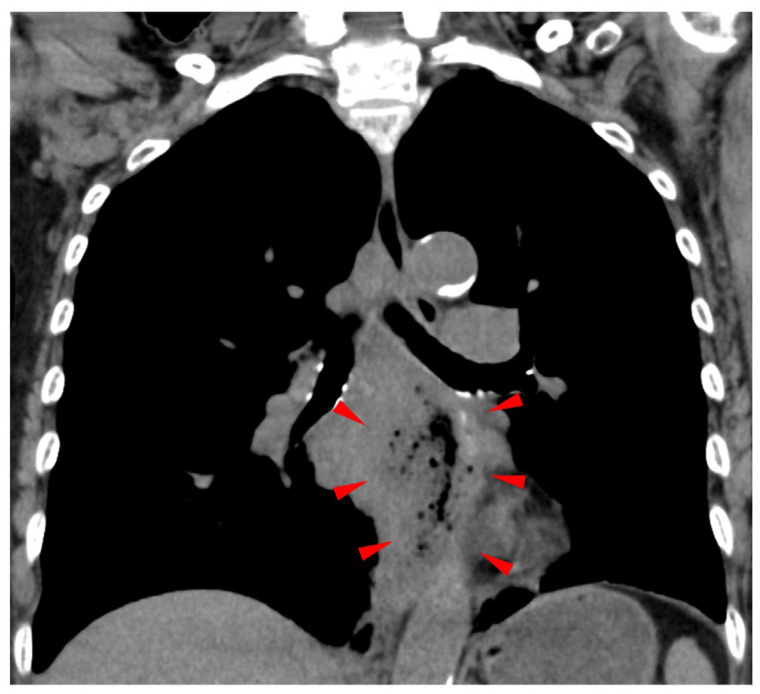
Mediastinal abscess caused by a fistula. A CT scan showed a mediastinal abscess resulting from an esophago-tracheobronchial fistula (indicated by arrowheads).

**Figure 2 jcm-13-07382-f002:**
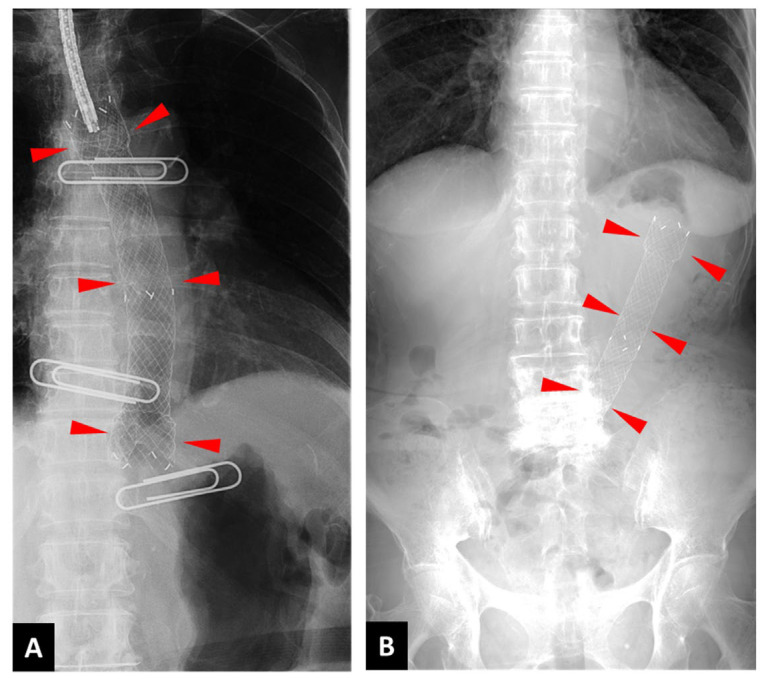
X-ray images of the implanted metallic stent. (**A**) A fully covered metallic stent was successfully placed in the esophagus (indicated by arrowheads). (**B**) The stent unexpectedly migrated into the stomach (indicated by arrowheads).

**Figure 3 jcm-13-07382-f003:**
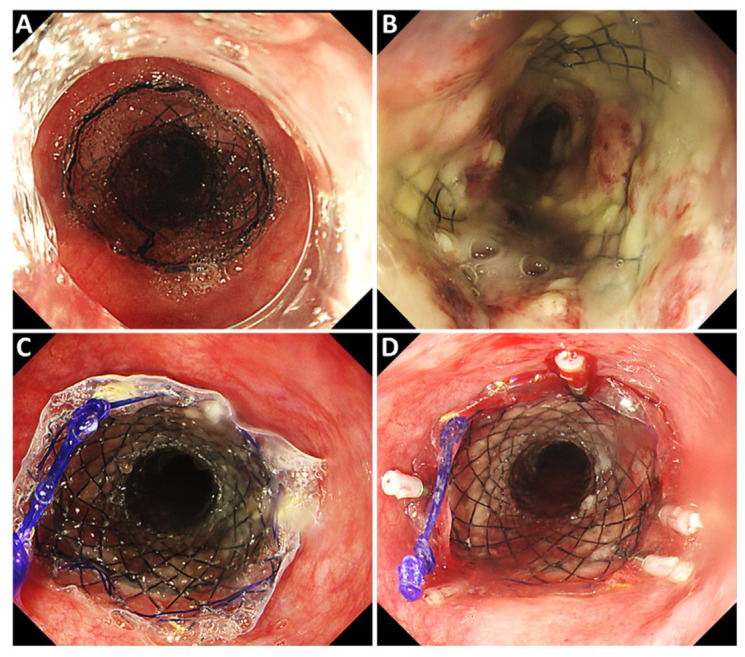
Procedure for the double-stent method. (**A**) An uncovered metallic stent was placed in the esophagus. (**B**) A few days later, the stent was securely fixed by the tumor tissue growing inside it. (**C**) A fully covered metallic stent of the same size was placed inside the uncovered stent (double-stent configuration). (**D**) Both stents were secured endoscopically with hemostatic clips.

**Table 1 jcm-13-07382-t001:** Gastrointestinal stent types and features.

Stent Type	Merit	Demerit
Uncovered	Hardly deviate	Frequent restenosis due to in-stent growth of tumor or mucosal hyperplasia Hard to remove
Partially covered	No in-stent growth of tumor in the covered area	Common restenosis due to mucosal hyperplasia at the flare Possibly deviate
Fully covered	No restenosis due to in-stent growth of tumor or mucosal hyperplasia Easy to remove	Easily deviate

## Data Availability

Data related to this case that are not included in the manuscript are available from the corresponding author upon reasonable request.

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
