# Peer review of "Successful Treatment of an Esophago-Tracheobronchial Fistula Using Double Stenting to Correct Initial Stent Migration: A Case Report and Literature Review"

_jcm, 2024, doi:10.3390/jcm13237382_

Round 1

Reviewer 1 Report

Comments and Suggestions for Authors

The study presents an innovative double stenting technique aimed at overcoming the issue of stent migration in treating esophago-tracheobronchial fistulas in advanced esophageal cancer. It describes the use of a combination of uncovered and fully covered metal stents to effectively prevent stent migration.

This well-documented case report offers a promising foundation for future clinical studies. By expanding the sample size, it will be possible to more thoroughly validate the efficacy and safety of the double stenting approach across diverse patient populations.

The present case report by Yuya et al. addresses the clinical management and treatment of an elderly woman diagnosed with advanced thoracic esophageal cancer, focusing on the complications associated with esophageal stenting and the effectiveness of a double stenting technique for managing fistulas.

The topic is both original and relevant, as it explores innovative approaches to managing complications of advanced esophageal cancer. It effectively prevents stent migration, reducing the risks associated with patient treatment. Additionally, the authors discuss that conducting comparative studies of different stenting techniques and provide valuable insights into the effectiveness of each method and related complications.

Specific Comments:

1. The authors should consider providing more detailed follow-up data on the patient’s nutritional status and any long-term outcomes following her discharge.

2. Regarding Figure 3: If the stent is securely fixed to the tumor, is there still a perceived risk of stent migration? Is it sufficient to use only one uncovered stent, and is there any testing to support this view?

Author Response

Reviewer 1

The authors should consider providing more detailed follow-up data on the patient’s nutritional status and any long-term outcomes following her discharge.

I really appreciate you for having reviewed my article. Based on your appropriate suggestion, I added the following sentences “The quantity and content of the diet was gradually increased, and on the 40th day she was able to consume a normal diet. With the increase in food intake, her body weight increased from 42 kg (on admission) to 45.5 kg and nutritional status improved.” in the “Case presentation” section.

Regarding Figure 3: If the stent is securely fixed to the tumor, is there still a perceived risk of stent migration? Is it sufficient to use only one uncovered stent, and is there any testing to support this view?

Thank you for your comments. As you noted, the uncovered stent is firmly anchored to the surface of the tumor, so the risk of deviation is very low. In this state, however, it is impossible to deter infection because the fistula is not completely covered. That was why a double stent technique was performed with a full-coverage stent.

Reviewer 2 Report

Comments and Suggestions for Authors

1)      There is an important reference that should be cited.1

2)      The manuscript is relatively well written, but there are some errors in grammar and syntax.  For example, the use of the term “drawbacks” seems colloquial.  Perhaps the authors could choose a different word, for example “limitations?”  Instead of describing the patient as “an elderly woman in her 80s,” would it be better to just state the patient's age?  The paper would benefit from copyediting.

3)      The authors indicate that TEF occurs into 5-15% of patients with esophageal cancer.  This seems high.  Please provide a citation to support this.

4)      The authors list several options for management of TEF.  Surgery is not discussed here.

5)      Please clarify in the beginning of the manuscript that esophageal stents are being used and not tracheal stents.  Is there ever a role for tracheal stents?

6)      As indicated by the authors, the technique for esophageal stenting to manage TEF requires adequate overlap and anchoring.  This may be difficult in patients with proximally located tumors who can experience a globus sensation or inadequate proximal purchase due to the location of the cricopharyngeus.  It would be helpful if the authors could report described locations for TEF’s, including their relative frequencies.  I believe that these lesions may be commonly located in the proximal esophagus, usually associated with squamous cell carcinoma, and may not always be amenable to stenting.

7)      This CT images are suggestive of a mediastinal perforation.  How was the diagnosis of TEF confirmed?  Did the patient undergo contrast-enhanced imaging, for example with an esophagram?

8)      This lesion was located in the mid-esophagus.  Why not just use a partially covered stent in the 1st place?  Was the fully covered stent that was initially used anchored with Endoclips?

9)      Why did the authors wait 20 days to start an oral diet?  Was an esophagram performed after the 2nd stenting procedure in order to confirm resolution of the fistula/leak?

10)   The patient presented with dysphagia.  The authors indicate that stent migration is rare in patients who have a stenosis.  What was the degree of stenosis identified at the time of initial endoscopy?

11)   The authors indicate that two patients have been treated with the “double stenting” technique.  Was the 2nd case described somewhere else?  Why not present both cases here?

12)   The authors indicate that stent costs of $745 and $845 are expensive.  What is the cost of a 43-day hospitalization in Japan?

 1.        Spivak H, Katariya K, Lo AY, Harvey JC. Malignant tracheo-esophageal fistula: use of esophageal endoprosthesis. J Surg Oncol. 1996;63:65-70.

Author Response

Reviewer 2

There is an important reference that should be cited.

Spivak H, Katariya K, Lo AY, Harvey JC. Malignant tracheo-esophageal fistula: use of esophageal endoprosthesis. J Surg Oncol. 1996; 63:65-70.

I really appreciate you for having reviewed my article and thank you for your appropriate suggestion. I cited the important paper in the main text as [3].

The manuscript is relatively well written, but there are some errors in grammar and syntax.  For example, the use of the term “drawbacks” seems colloquial.  Perhaps the authors could choose a different word, for example “limitations?”  Instead of describing the patient as “an elderly woman in her 80s,” would it be better to just state the patient's age?  The paper would benefit from copyediting.

Thank you for your comments. I replaced the word “drawbacks” into “limitations” and rewrote the sentence of the presented case as the following “An 82-year-old woman”.

The authors indicate that TEF occurs into 5-15% of patients with esophageal cancer.  This seems high.  Please provide a citation to support this.

Thank you for your important comment. I prepared the two articles below and cited these in the main text as [1,2].

  1. Tsushima T, Mizusawa J, Sudo K, et al., Risk factors for esophageal fistula associated with chemoradiotherapy for locally advanced unresectable esophageal cancer: a supplementary analysis of JCOG0303. Medicine (Baltimore). 2016; 95: e3699.
  2. Hihara J, Hamai Y, Emi M, et al., Role of definitive chemoradiotherapy using docetaxel and 5-fluorouracil in patients with unresectable locally advanced esophageal squamous cell carcinoma: a phase II study. Dis. Esophagus. 2016; 29, 1115–20.

The authors list several options for management of TEF.  Surgery is not discussed here.

Thank you for your valuable comment. Usually, cases of fistula formation are classified as advanced stage, and the general condition of the patient is often poor. In these cases, the esophagus is tightly adhered to the bronchus and lungs due to tumor invasion, making it difficult to close the fistula by surgical dissection, which is a physically invasive procedure.

I added the following sentences “Of course, except endoscopic procedure, surgical approach might be considered as a treatment method of esophago-tracheobronchial fistulas. However, cases of fistula formation are basically classified as advanced stage, and the general condition of these patients is often poor. In these cases, the esophagus is tightly adhered to the bronchus and lungs due to tumor invasion, making it difficult to close the fistula by surgical dissection, which is a physically invasive procedure.” in the “Discussion” section.

Please clarify in the beginning of the manuscript that esophageal stents are being used and not tracheal stents.  Is there ever a role for tracheal stents?

Thank you for your appropriate comment. When esophagotracheal fistulas occur, they usually perforate not only the trachea but also the mediastinum and other organs. Even if there is only perforation of the trachea at the time of stent placement, other perforations are likely to occur as the disease progresses, so we put a covered stent to cover the entire esophageal tumor area when we place the stent. For this reason, we prefer to place the stent in the esophagus rather than in the trachea.

As indicated by the authors, the technique for esophageal stenting to manage TEF requires adequate overlap and anchoring.  This may be difficult in patients with proximally located tumors who can experience a globus sensation or inadequate proximal purchase due to the location of the cricopharyngeus.  It would be helpful if the authors could report described locations for TEF’s, including their relative frequencies.  I believe that these lesions may be commonly located in the proximal esophagus, usually associated with squamous cell carcinoma, and may not always be amenable to stenting.

Thank you for your important comment. In an epidemiological study in Asia, where squamous cell carcinoma is common, the site-specific incidence of 221 cases of esophageal fistula was 31.7% in the upper part, 41.6% in the middle part, and 26.7% in the lower part [Xin Guan, Chao Liu, Tianshuo Zhou et al., Survival and prognostic factors of patients with esophageal fistula in advanced esophageal squamous cell carcinoma.Biosci Rep. 2020 Jan 14;40(1):BSR20193379.]. In a Canadian study report, where adenocarcinoma is relatively common, the incidence of esophageal cancer by site was 8.4% in the upper part, 22.7% in the middle part, and 68.9% in the lower part [Michael C Otterstatter, James D Brierley, Prithwish De et al., Esophageal cancer in Canada: trends according to morphology and anatomical location. Can J Gastroenterol. 2012 Oct;26(10):723-7.]. In other words, the frequency of cancer in the upper esophagus, as well as the frequency of fistulas, is generally not high. Even if a fistula develops in the upper esophagus, there have been reports of successful closure using the novel proximal release-type stent (HANARO-STENT). [Ueda T, et al., Clin Gastroenterol Hepatol. 2024;22:A28-9.]

This CT images are suggestive of a mediastinal perforation.  How was the diagnosis of TEF confirmed?  Did the patient undergo contrast-enhanced imaging, for example with an esophagram?

Thank you for your appropriate comment. Esophagography to prove esophagotracheobronchial fistula was not initially performed because it could exacerbate fistula infection. As you indicated, I believe that in this case the trachea as well as the mediastinum were perforated. I searched PubMed for the term esophago-tracheobronchialmediastinum fistula to describe this situation. However, not a single reference used such a term, so I used esophago-tracheobronchial fistula.

This lesion was located in the mid-esophagus.  Why not just use a partially covered stent in the 1st place?  Was the fully covered stent that was initially used anchored with Endoclips?

Thank you for your valuable comment. A review of cover stents found no difference in deviation rates between full and partial cover types [Wang C, et al. BMC Canc, 2020; 20: 73.]. There are cases in which stent placement requires urgent removal due to pressure drainage of other organs by opening of the stent, and we have experienced urgent removal of stents in cases of tracheal pressure drainage. Therefore, our policy is to first select a full-coverage stent that can be removed urgently if the deviation rate is the same.

Why did the authors wait 20 days to start an oral diet?  Was an esophagram performed after the 2nd stenting procedure in order to confirm resolution of the fistula/leak?

Thank you for your comment. The fistula was closed with the placement of a cover stent, but the pneumonia and mediastinitis that had developed before the stent was placed were quite inflammatory and were being treated by changing the antibiotic from a cephem antibiotic to a carbapenem antibiotic. The patient's general condition was very critical, and the first priority was to avoid another exacerbation of the infection, so the patient was fasted until the inflammation subsided.

The patient presented with dysphagia.  The authors indicate that stent migration is rare in patients who have a stenosis.  What was the degree of stenosis identified at the time of initial endoscopy?

Thank you for your valuable comment. Usually, if the lumen can hold an oral scope (ø9.9 mm), the patient is considered to be in a minimum condition for oral intake. Therefore, if the scope can be passed through, there is no stenosis; if not, there is stenosis. In both of the two cases presented here, passage of the scope was possible.

The authors indicate that two patients have been treated with the “double stenting” technique.  Was the 2nd case described somewhere else?  Why not present both cases here?

Thank you for your appropriate comment. We added the case presentation of the second patient. Please see the “Case presentation” section. The added sentences were the followings; Case 2. A 71-year-old man visited our hospital after experiencing discomfort upon swallowing for about a month. He had no medical history, but had a beer drinking history of one liter every day and smoking history (20 pieces per day for 30 years). Upper gastrointestinal endoscopy revealed advanced esophageal squamous cell carcinoma (cT3N2M0 cStage III) in the mid-thoracic esophagus (30–35 cm from the incisors), and neoadjuvant DCF chemotherapy (docetaxel, cisplatin, and 5-fluorouracil) was performed. During the course of treatment, the patient became aware of fever and persistent cough, and the inflammatory findings of blood tests and CT of the chest were most suspicious for a mediastinal abscess associated with esophageal perforation (Suppl. Fig. 1). After admission, he was fasting and receiving antimicrobials, and a full-coverage esophageal stent (HA-NAROSTENT®) with an outer diameter (OD) of 18 mm and a length of 12 cm was implanted against the tumor area for fistula closure on the third day, but it was found that the stent had fallen out in the stomach on the ninth day (Suppl. Fig. 2). To avoid the possibility of stent redislodgement, an uncovered stent (Niti-S®), 18 mm OD and 12 cm long, was placed in the tumor with fistula on the 16th day, and a covered stent (Niti-S®) of the same size was placed inside it one week later to create a complete double-layer structure, The mouth ends of both stents were fixed with clips (Suppl. Fig. 3). Esophagogram with Gastrografin® contrast on the 26th day showed no contrast leak (Suppl. Fig. 4) and oral in-take was resumed. Radiotherapy and chemotherapy for esophageal cancer were started on the 31st and 43rd day, respectively, but there was no stent deviation or worsening of mediastinal abscess, and the patient was discharged on the 48th day. Thereafter, there were no stent deviations or occlusions during the 216 days until death.

Please see the supplementary figure legends, too.

Supplementary figure 1: Chest CT images. Arrow heads indicate mediastinal abscess caused by a fistula in an axial (left) and coronal (right) images.

Supplementary figure 2: Abdominal X-ray image. Arrow heads indicate the dislocated stent in his stomach.

Supplementary figure 3: Endoscopic images of the double stenting technique. Just after implanting an uncovered stent (A), a few days later (B), after implanting the fully covered stent in the inside of the previous uncovered stent (C), and just after completing ligation with clips (D).

Supplementary figure 4: Esophagogram with Gastrografin®.

The authors indicate that stent costs of $745 and $845 are expensive.  What is the cost of a 43-day hospitalization in Japan?

Thank you for your comment. A typical hospital stay for 43 days for radiation chemotherapy for esophageal cancer costs approximately USD 16,000. For this stenting, (450USD for stenting and 850USD for materials) x 2 times will be added. In Japan, patients bear 30% of the above amount (10% for those over 75 years of age), and since the cost of stenting affects the patient's share, we believe that the fact that the stent and implantation procedure cost twice is a problematic aspect of this method.